# A Naked Lyophilized mRNA Vaccine Against Seasonal Influenza, Administered by Jet Injection, Provides a Robust Response in Immunized Mice

**DOI:** 10.3390/vaccines14010056

**Published:** 2026-01-02

**Authors:** Sergei V. Sharabrin, Svetlana I. Krasnikova, Denis N. Kisakov, Mariya B. Borgoyakova, Vladimir A. Yakovlev, Elena V. Tigeeva, Ekaterina V. Starostina, Victoria R. Litvinova, Lyubov A. Kisakova, Danil I. Vahitov, Kristina P. Makarova, Ekaterina A. Volosnikova, Ksenia I. Ivanova, Alexander A. Bondar, Nadezhda B. Rudometova, Andrey P. Rudometov, Alexander A. Ilyichev, Larisa I. Karpenko

**Affiliations:** 1State Research Center of Virology and Biotechnology “Vector”, Rospotrebnadzor, Koltsovo 630559, Russia; 2Institute of Chemical Biology and Fundamental Medicine, Siberian Branch, Russian Academy of Sciences, Novosibirsk 630090, Russia

**Keywords:** mRNA-vaccine, seasonal influenza vaccine, mRNA delivery, lipid nanoparticles, jet injection

## Abstract

**Background**: Seasonal influenza remains a significant public health problem, and the constant antigenic drift of viruses requires regular vaccine updates. mRNA vaccines offer a promising platform for the development of new, effective influenza vaccines. Administration of the naked mRNA vaccine using a needle-free jet injection system further enhances its safety, reduces cost, and eliminates the need for lipid nanoparticles, which are traditionally used for mRNA delivery. Lyophilization of naked mRNA allows for long-term storage at +4 °C. **Methods**: We designed and produced an mRNA vaccine against seasonal influenza, designated mRNA-Vector-Flu, encoding the hemagglutinin (HA) of the A/Wisconsin/67/2022(H1N1)pdm09, A/Darwin/9/2021(H3N2), and B/Austria/1359417/2021 strains. The vaccine was lyophilized and stored for 1 month in a refrigerator (+4 °C). A comparative immunogenicity study was conducted between synthesized immediately before use prepared and lyophilized naked mRNA-Vector-Flu. The preparations were administered to BALB/c mice using a jet needleless injection twice, 3 weeks apart. Immunogenicity was assessed on day 35 of the study. **Results**: A comparative immunogenicity study of naked mRNA-Vector-Flu demonstrated that both the synthesized immediately before use prepared formulation and the lyophilized form, stored at +4 °C for a month, induced similar levels of virus-specific antibodies and generated a pronounced T-cell immune response. **Conclusions**: Delivery of the naked mRNA vaccine using a needle-free jet injection ensures a high-level immune response, which improves its safety, reduces its cost, and eliminates the need for lipid nanoparticles traditionally used for mRNA delivery. At the same time, lyophilization of the naked mRNA vaccine preserves its biological activity and ensures its storage for at least a month at +4 °C temperatures. Our results demonstrate that our proposed approach can be considered a promising direction for the development and improvement of the mRNA vaccine platform.

## 1. Introduction

Seasonal influenza remains a significant public health problem, with over 3 million influenza-related hospitalizations worldwide annually [1,2] and up to half a million deaths from this disease recorded annually [3]. Vaccination against seasonal influenza is an effective means of eliciting immunity, helping to reduce the significant burden of annual influenza epidemics [4,5].

Various types of influenza vaccines exist, including live attenuated, inactivated (whole-virion, split, subunit), and recombinant vaccines [6,7,8]. However, the constant antigenic drift of circulating influenza viruses renders seasonal influenza vaccines ineffective, necessitating annual reformulated of vaccines. In this case, mRNA-based vaccine platforms offer advantages over standard influenza vaccine technologies. The low cost of producing mRNA vaccines compared to conventional split subunit vaccines developed using RNA cells and the ability to quickly and easily replace the target gene in mRNA vaccines without changing the production technology itself allows for a timely response to the emergence of new virus strains [9]. mRNA vaccine technology has enabled the rapid creation of effective and safe drugs for the prevention of coronavirus. In total, approximately one billion people worldwide have been vaccinated with mRNA-based SARS-CoV-2 vaccines, and their effectiveness has been demonstrated [10,11]. The production of mRNA vaccines does not require the use of chicken embryos, which avoids mutations in the immunogen sequence during the production of vaccine batches. mRNA vaccines do not induce an unwanted immune response in the recipient, as is typical with vector vaccines, and therefore can be administered multiple times. Another important advantage is the efficient activation of both humoral and cellular virus-specific immunity, since the target protein is expressed endogenously from mRNA [12,13].

Currently, several teams are developing mRNA vaccines against seasonal influenza [14,15,16], and several vaccines, including Moderna’s mRNA-1010, are undergoing clinical trials [17,18,19,20]. Lipid nanoparticles are typically used to deliver mRNA vaccines. Lipid nanoparticles consist of four lipid components that ensure effective encapsulation and release of mRNA, prolonged circulation in the body, and complex stability [21]. However, certain limitations have been identified, related to both the complexity of storage (down to −80 °C) and, consequently, the logistics of such vaccines, and adverse post-vaccination effects. These effects are believed to be largely related to the nature of the lipids coating the mRNA [22,23].

Therefore, alternative delivery methods have recently attracted increasing attention. Thus, the recently published [24,25,26,27] jet injection (JI) method is a promising alternative to LNPs for mRNA delivery. Jet injection is a physical delivery method in which vaccines and other therapeutic drugs are administered in fractions of a second using a high-speed jet through a nozzle under high pressure. This allows for efficient drug delivery intradermally, intramuscularly, or subcutaneously, without the need for a needle [28]. The most attractive feature of this method is that the mRNA vaccine is administered as a naked molecule, completely eliminating the negative effects of the lipid components encapsulating the mRNA molecule. However, in this case, the issue of storage and stability of naked mRNA molecule preparations becomes relevant, so we decided to use lyophilization.

The aim of this study was to compare the immunogenicity of the mRNA-Vector-Flu trivalent mRNA vaccine against seasonal influenza virus, synthesized immediately before use and its lyophilized form, administered via jet injection.

## 2. Materials and Methods

### 2.1. Bacterial and Viral Strains, Cell Cultures, and Plasmids

The *E. coli* strain NEB Stable (New England Biolabs Inc., Ipswich, MA, USA) was used for genetic engineering work. For in vitro analysis of mRNA functionality, the continuous human embryonic kidney cell line HEK293, provided by the Microorganism Collection Department of the State Research Center of Virology and Biotechnology Vector (Rospotrebnadzor, Koltsovo, Russia), was used.

To obtain DNA templates for mRNA synthesis, plasmids encoding the hemagglutinins of the corresponding influenza virus strains were used: pVAX-H1-24 encoding the hemagglutinin of the influenza virus A/Wisconsin/67/2022(H1N1)pdm09, pVAX-H3-24 encoding the hemagglutinin of the influenza virus A/Darwin/9/2021(H3N2) and pVAX-HB-24 encoding the hemagglutinin of the influenza virus B/Austria/1359417/2021 (FBRI SRC VB Vector, Rospotrebnadzor). To obtain mRNA-GFP, the previously obtained pVAX-C3-GFP matrix (FBRI SRC VB “Vector”, Rospotrebnadzor) was used.

### 2.2. Templates for mRNA Synthesis

Hemagglutinins of influenza viruses recommended by WHO for the Northern Hemisphere for the 2023–2024 season [29] were used as antigens for the mRNA vaccine: A/Wisconsin/67/2022(H1N1)pdm09 (EPI_ISL_15928538), A/Darwin/9/2021(H3N2) (EPI_ISL_20142977), B/Austria/1359417/2021 (EPI_ISL_983345) without the transmembrane and cytoplasmic domains. To stabilize the trimeric structure of hemagglutinins, a T4 trimerizing domain was added to the C-terminus [30,31]. To maintain the uncleaved form of the rHA/H1 and rHA/H3 proteins, the following amino acid substitutions were introduced into the pH switch region of the HA2 subunit: H26 and H106, where histidine was replaced by tryptophan (W) and arginine (R), and amino acids K51 and E103 were replaced by isoleucine [32]. The codon composition of the sequence was optimized using the codon adaptation tool (https://www.jcat.de/, accessed 1 September 2025). To obtain DNA templates for mRNA synthesis, the previously developed pVAX-C3-PolyA DNA cassette [33], into which the hemagglutinin genes were cloned, was used. Cassette pVAX-C3-PolyA contains a T7 promoter that modified for efficient incorporation of the AG-Cap analogue during mRNA synthesis, 5′-UTR of ChM and 3′-UTR of human β-globin, and a 100-nucleotide poly(A) tail. The source of the hemagglutinin genes were the plasmids pVAX-H1-24, pVAX-H3-24 and pVAX-HB-24, encoding the corresponding hemagglutinin sequence. To obtain DNA templates for mRNA synthesis, routine cloning was performed: the pVAX-C3-PolyA cassette and the pVAX-H1\3\B plasmids were treated with endonucleases CciNI and BamHI (SibEnzyme, Novosibirsk, Russia). Then, the resulting DNA fragments were ligated using T4 DNA ligase (SibEnzyme, Novosibirsk, Russia), and the resulting mixture was used to transform *E. coli* cells. The resulting constructs were named pVAX-C3-H1-24, pVAX-C3-H3-24, and pVAX-C3-HB-24. The DNA sequences of the resulting plasmids were confirmed by restriction analysis and Sanger sequencing.

DNA templates were purified using the HiPure Plasmid Mini Kit (Catalog No.: P100103, Magen, Guangzhou, China) and linearized using the restriction endonuclease Bso31I (SibEnzyme, Novosibirsk, Russia).

### 2.3. In Vitro mRNA Synthesis

mRNA was synthesized using linearized pVAX-C3-H1-24, pVAX-C3-H3-24, pVAX-C3-HB-24, and pVAX-C3-GFP plasmid templates together with a commercial in vitro transcription kit (Yeasen, Shanghai, China). Each reaction mixture contained 1 μg of linearized DNA, T7 RNA polymerase with its corresponding buffer, the AG-Cap structure analog (m7GmAmG cap analog, Biolabmix, Novosibirsk, Russia; 10 mM), a ribonucleotide triphosphate mix (10 mM) in which uridine was substituted with N1-methylpseudouridine (Biolabmix, Novosibirsk, Russia), RNase inhibitor (BelBioLab, Moscow, Russia), and nuclease-free water. The synthesis procedure, previously described by our group in [33], involved enzymatic transcription, DNase treatment to remove template DNA, and subsequent purification steps. The resulting transcripts were designated as mRNA-H1, mRNA-H3, mRNA-HB, and mRNA-GFP.

### 2.4. PCR to Determine the Presence of Template DNA Residues in the mRNA Preparation

For PCR, primers complementary to the 5′- and 3′-UTR sequences of the mRNA encoded in the template DNA were used. The BioMaster HS-Taq PCR kit (2×) (Biolabmix, Novosibirsk, Russia), primers, and 1 μg of synthesized mRNA were used. Linearized template DNA was used as a control.

### 2.5. Capillary Electrophoresis

The purity, uniformity, and molecular size of the synthetic mRNAs were evaluated using an Agilent 2100 BioAnalyser (Agilent Technologies, Santa Clara, CA, USA). Analysis was performed by microcapillary electrophoresis with the Agilent RNA 6000 Pico kit (Agilent Technologies, Vilnius, Lithuania) following the manufacturer’s instructions.

### 2.6. Encapsulation of mRNA in Lipid Nanoparticles

Encapsulation of mRNA into lipid nanoparticles and subsequent characterization were carried out following the procedure described by Yakovlev et al., 2025 [34]. Phase mixing was performed using an automated nanoparticle production system (Dolomite Microfluidics, Royston, UK) equipped with a hexagonal herringbone micromixer chip (Dolomite Microfluidics, Royston, UK). The aqueous phase contained mRNA dissolved in 100 mM citrate buffer (pH 4). The ethanol phase consisted of a lipid mixture composed of ionizable lipids, phospholipids, helper lipids, and PEG-lipids at molar ratios of 50:10:38.5:1.5, respectively; all lipid components were solubilized in 96% ethanol. The lipid formulation included the ionizable lipid SM-102 (heptadecan-9-yl 8-((2-hydroxyethyl)(6-oxo-6-(undecyloxy)hexyl)amino)octanoate), the phospholipid DSPC (1,2-distearoyl-sn-glycero-3-phosphocholine), cholesterol as a helper lipid, and the PEG-lipid DMG-PEG2000 (1-monomethoxypolyethylene glycol-2,3-dimyristylglycerol with PEG of average molecular weight 2000).

The resulting lipid nanoparticles were analyzed using dynamic light scattering (DLS). DLS measurements confirmed that the mRNA-LNP formulations formed monodisperse nanoparticle suspensions, with PdI values of 0.158 ± 0.013 for mRNA-H1, 0.178 ± 0.006 for mRNA-H3, and 0.239 ± 0.034 for mRNA-HB. The mean hydrodynamic diameters of the LNPs were 94.42 ± 2.48 nm, 95.34 ± 0.38 nm, and 97.56 ± 2.44 nm for the respective formulations. The ζ-potentials measured across three series were 0.35 ± 0.12 mV, 0.54 ± 0.42 mV, and 0.05 ± 0.62 mV, values which were consistent with theoretical predictions. mRNA encapsulation efficiency, assessed using the Quant-iT RiboGreen assay (Life Technologies, Waltham, MA, USA), exceeded 90% for all formulations.

### 2.7. mRNA Lyophilization

Lyophilization of mRNA sample solutions was performed in an ED-DF21A-K freeze-drying chamber (ERSTVAK, Moscow, Russia) in automatic mode with a pneumatic capping option. Three cryoprotectant variants were used for lyophilization: the first was based on sucrose, the second was based on mannose, and the third was based on trehalose. The sterile mRNA preparation with one of three cryoprotectants was dispensed into 0.5 mL vials of 3 mL and lyophilized for 24 h. The lyophilization process was carried out under standard conditions: freezing samples at −50 ± 2 °C for 8 h, and lyophilizing samples at 22 ± 2 °C for the remainder of the time. The lyophilized mass was a white tablet. The dried mRNA was stored at temperatures of −20 °C, +4 °C, and +20 °C.

### 2.8. Immunization of BALB/c Mice

Animal experiments were carried out in compliance with the Guide for the Care and Use of Laboratory Animals. All procedures were approved by the Laboratory Animal Care and Use Committee of the Federal Research Center of Virology and Biotechnology “Vector”, Rospotrebnadzor (Bioethics Committee Protocol No. 3, 2024). Mice were housed under a 12 h light/dark cycle with unrestricted access to food and water.

The study utilized inbred BALB/c mice weighing 16–18 g at the beginning of the experiment. Preparations of mRNA-Vector-Flu-LNP in 100 μL of PBS were injected intramuscularly into the quadriceps of the left hind limb using insulin syringes equipped with a 29G needle. The mRNA-H1, mRNA-H3, mRNA-HB, and mRNA-Vector-Flu formulations, dissolved in 50 μL of saline, were administered into the quadriceps by the IM with Jet-injection as described earlier [27,33]. Jet-injection immunization was performed using a Comfort-IN needle-free jet injector (MIKA MEDICAL CO, Busan, Republic of Korea) with the following parameters: injection velocity of 220 m/s, pressure of 6.5 bar, and injection duration of 0.33 s. Disposable nozzles were used for each application. The commercial seasonal influenza vaccine Flu-M (FSUE SPbNIIVS FMBA of Russia, 2025) was administered intramuscularly into the quadriceps using a syringe.

The experiment consisted of three phases.

Phase 1: A comparative evaluation of the immunogenicity of the developed mRNA vaccine. Mice were divided into seven groups (*n* = 6 per group) and vaccinated on days 0 and 21. Group 1 received 30 μg mRNA-H1; Group 2–30 μg mRNA-H3; Group 3–30 μg mRNA-HB; Group 4—the trivalent mRNA-Vector-Flu vaccine (30 μg of each component, 90 μg total); Group 5—the trivalent mRNA-Vector-Flu-LNP vaccine (10 μg per component, 30 μg total); Group 6—250 μL of the Flu-M commercial vaccine; Group 7 was the unimmunized control. Jet injection was used for groups 1–4, whereas groups 5 and 6 received injections by syringe. The 30 μg mRNA dose was selected based on findings from earlier work [27,33] and studies on jet-injection performed by other researchers [26]. On day 35, blood was collected from the retro-orbital sinus, and animals were euthanized by cervical dislocation.

Phase 2: Evaluation of dose-dependent effects of the mRNA vaccine. Mice were divided into three groups (*n* = 6 each) and immunized on days 0 and 21. Group 1 received the trivalent mRNA-Vector-Flu formulation at 10 μg per mRNA (30 μg total); Group 2 received 30 μg per mRNA (90 μg total); Group 3 received 50 μg per mRNA (150 μg total). Immunizations were performed using jet injection. Blood was collected on day 35 from the retro-orbital sinus for humoral immunity assessment.

Phase 3: Assessment of the effectiveness of lyophilized mRNA vaccine formulations. Two groups of six mice each were immunized on days 0 and 21. Group 1 received the lyophilized trivalent mRNA-Vector-Flu vaccine containing 15 μg of each component (45 μg total). Group 2 received the non-lyophilized mRNA-Vector-Flu formulation with the same component doses. Animals were vaccinated using jet injection. On day 35, retro-orbital blood samples were obtained for humoral response evaluation, including HAI and microneutralization assays. For assessment of T-cell responses, mice were euthanized by cervical dislocation and their spleens were collected.

### 2.9. Enzyme-Linked Immunosorbent Assay (ELISA)

The enzyme-linked immunosorbent assay (ELISA) was carried out as previously reported [35]. Recombinant eukaryotic hemagglutinin proteins H1 (A/Wisconsin/67/2022(H1N1)pdm09), H3 (A/Darwin/9/2021(H3N2)), and HB (B/Austria/1359417/2021), produced earlier at the Bioengineering Department of the State Research Center of Virology and Biotechnology Vector, Rospotrebnadzor, served as antigenic targets. Rabbit anti-mouse IgG antibodies conjugated to horseradish peroxidase (Sigma-Aldrich, St. Louis, MO, USA) were employed as secondary antibodies. TMB (Amresco LLC, Solon, Ohio, USA) was used as the chromogenic substrate. Following incubation, the enzymatic reaction was terminated with a stop reagent (1 M hydrochloric acid), and absorbance was recorded at 450 nm using a Varioskan Lux microplate reader (Thermo Fisher Scientific, Waltham, MA, USA).

### 2.10. Evaluation of the Cellular Immune Response by ELISpot

Splenocytes were obtained by gently dissociating individual spleens through nylon strainers with pore sizes of 70 and 40 μm (BD Falcon™, Franklin Lakes, NJ, USA). Erythrocytes were eliminated using a red blood cell lysis buffer (Sigma, Virginia Beach, VA, USA). The magnitude of the T-cell response in immunized mice was assessed by quantifying the number of splenocytes secreting IFN-γ using an IFN-γ ELISpot assay. The procedure was carried out with IFN-γ ELISpot kits (MABTECH, Nacka Strand, Sweden) following the manufacturer’s protocol. For cell stimulation, a pool of synthetic peptides (10–20 aa in length) representing conserved T-cell epitopes of the hemagglutinin proteins from influenza viruses A/Wisconsin/67/2022 (H1N1)pdm09, A/Darwin/9/2021 (H3N2), and B/Austria/1359417/2021 was used; peptides were produced by AtaGenix Laboratories (Wuhan, China). Each peptide was applied at a concentration of 20 μg/mL. The number of IFN-γ–producing cells was determined using an ELISpot reader (Carl Zeiss, Oberkochen, Germany).

### 2.11. In Vitro Microneutralization Assay

The in vitro neutralization assay was performed with influenza viruses A/Buryatia/106-6V/2022 (H1N1), A/Darwin/9/2021 (H3N2), and B/Austria/1359417/2021 as described in study [36], with modifications. The difference lay in the method used to visualize the final result: two days after infection, the cells were stained with a crystal violet solution (1.3 g of dye dissolved in 50 mL of 96% ethanol, brought to 700 mL with distilled water, and supplemented with 300 mL of 40% formalin). The results were analyzed using an Agilent BioTek Cytation 5 multimode cell imaging reader (Thermo Fisher Scientific). The neutralization titer was defined as the highest serum dilution at which ≥50% virus neutralization was achieved, corresponding to ≥50% viable cells.

### 2.12. Hemagglutinin Inhibition (HAI) Assay

HAI assay was performed according to the HAI protocol based on World Health Organization (WHO) guidelines. Prior to the HI test, the collected animal serum was treated with RDE (Denka Seiken, Tokyo, Japan) for 18–20 h according to the manufacturer’s instructions to remove nonspecific thermostable inhibitors, then heated in a water bath at 56 °C to eliminate nonspecific thermolabile inhibitors and to inactivate the enzymatic activity of RDE. The final serum dilution of 1:10 was achieved by adding phosphate-buffered saline. The sera were tested in the HAI assay against 4 hemagglutinating units of antigen corresponding to the virus serotype. To determine the HAI titer, 0.5% turkey red blood cells and U-bottom plates were used. When calculating geometric mean titers, HAI values <1/10 were considered equal to 5. Inactivated β-propiolactone–treated influenza viruses A/Buryatia/106-6V/2022 (H1N1), A/Darwin/9/2021 (H3N2), and B/Austria/1359417/2021 were used as antigens.

### 2.13. Transfection of HEK293 Cells with mRNA

HEK293 cells were cultured in 24-well plates (Corning, New York, NY, USA) in DMEM medium (Sigma-Aldrich, St. Louis, MO, USA) supplemented with 10% FBS (HyClone, Logan, UT, USA) and 50 mg/mL gentamicin. When the cell monolayer reached approximately 70–80% confluence, it was transfected with mRNA encoding GFP using a PEI-based nucleic acid transfection kit for eukaryotic cells (BIOSPECIFICA, Novosibirsk, Russia). The transfection reagent was combined with 1 µg of mRNA, incubated for 15 min at room temperature, and subsequently applied to the cells. The plates were then placed in a CO_2_ incubator and incubated for 24 h.

### 2.14. Statistical Analysis

Statistical data processing was carried out using the GraphPad Prism 10.0 software package (GraphPad Software, San Diego, CA, USA). Quantitative variables were presented either as mean values with an interval or as a range from minimum to maximum. Nonparametric statistical methods were applied for their analysis. Differences between the study groups were evaluated using the nonparametric one-way Kruskal–Wallis analysis of variance with subsequent adjustment for multiple comparisons and Dunn’s hypothesis testing.

## 3. Results

### 3.1. Production of the Experimental Trivalent mRNA Influenza Vaccine mRNA-Vector-Flu

Previously, we developed a DNA template cassette for mRNA vaccine synthesis, called pVAX-C3-PolyA [33]. The modified chimeric β-globin sequence used in Moderna’s studies and the human β-globin sequence were included in the cassette as the 5′-UTR and 3′-UTR, respectively. These elements are necessary for increasing mRNA stability and translation efficiency. The initiator nucleotides GG in the T7 promoter were replaced with AG. This modification allows the use of the AG-Cap analog during in vitro transcription to create a “cap” at the 5′-end of mRNA, which is critical for its stability and translation. The plasmid contains a 100-nucleotide poly(A) tail containing an internal linker of 10 random nucleotides (e.g., 30(A)GCATATGACT70(A)). The poly(A) tail is important for mRNA stability and translation initiation. The poly(A) tail sequence ends with a Bso31I restriction endonuclease site. This element ensures that during DNA hydrolysis, the transcription template ends with an adenine, ensuring the precise termination of the synthesized mRNA.

The pVAX-H1-24 plasmid, carrying the hemagglutinin gene of the influenza A/Wisconsin/67/2022 (H1N1)pdm09 virus, was used as the source of the target gene for the trivalent mRNA components of the influenza vaccine. pVAX-H3-24, carrying the hemagglutinin gene of the influenza A/Darvin/9/2021 (H3N2) virus, and pVAX-HB-24, carrying the hemagglutinin gene of the influenza B/Austria/1359417/2021 virus (B/Victoria lineage). In all hemagglutinins, the transmembrane and cytoplasmic domains were removed, and the trimerizing domain of phage T4 was added to enhance immunogenicity (Figure 1a).

After inserting the hemagglutinin gene into the pVAX-C3-PolyA cassette, the pVAX-C3-H1-24, pVAX-C3-H3-24, and pVAX-C3-HB-24 constructs were obtained. These constructs were then used to synthesize the corresponding mRNAs.

### 3.2. mRNA Synthesis and Characterization

mRNA-C3-H1-24, mRNA-C3-H3-24, and mRNA-C3-HB-24 were synthesized using the method described in Section 2.3. The integrity and purity of the mRNA were verified by electrophoresis in a 2% agarose gel (Figure 1b) and capillary electrophoresis (Appendix A). The mRNA transcript size was expected to be approximately 1900 base pairs, consistent with the DNA template. The absence of a high-molecular-weight signal indicates complete removal of the DNA template from the preparation. PCR was also performed for confirmation (Figure 1c).

### 3.3. Immunogenicity Assessment of the mRNA-Vector-Flu Vaccine

To assess the immunogenicity of the seasonal influenza mRNA vaccine, BALB\c mice (*n* = 6) were immunized with each mRNA individually and in a mixture. A needle-free jet injection method was used for immunization, as described previously [33,34]. The mRNA dose was 30 μg of each immunogen per animal. As a control, one group of animals was immunized with the commercial Flu-M vaccine (SPbSRIVS FMBA of Russia) of the 2024–2025 formulation. Lipid nanoparticles, the gold standard of mRNA delivery, were also used as a control. Animals were immunized with a mixture of 10 μg of each mRNA encapsulated in LNPs, as described previously [25,27,34].

Animal sera were collected on day 35 of the experiment. To set up the ELISA, individual recombinant proteins of hemagglutinin H1, H3 and HB were used as the antigen, as described in Section 2.9. The results showed that all nonvariant’s of the trivalent mRNA vaccine against influenza mRNA-Vector-Flu generate an immune response against the corresponding antigen (Figure 2). The average titer for mRNA-H1 was 1:400,950, and for mRNA-H3 1:510,300, mRNA-HB 1:225,000. The commercial vaccine Flu-M showed less pronounced antibody titers, and in the case of hemagglutinin H3 the titer was practically equal to 0. The antibody titer in the group of animals immunized with the trivalent mRNA vaccine in LNP was slightly higher than in the group of animals immunized with jet injection, but the differences were not significant (Figure 2).

### 3.4. Comparison of the Dose-Dependent Effect of the Trivalent mRNA Vaccine

For a dose-dependent comparison, BALB/c mice were immunized twice with mRNA-Vector-Flu using JI, at doses of 10, 30, and 50 μg of each immunogen. Results showed that at a 10 μg dose of trivalent mRNA, the average titer ranged from 1:100,000 to 1:200,000, depending on the antigen in the ELISA (Figure 3). At a dose of 30 μg of each immunogen, the average titer was approximately 1:500,000. At a dose of 50 μg of each immunogen, we did not observe an increase in the humoral immune response compared to the 30 μg dose. This may be due to the high immunogen dose, which resulted in immunosuppression.

### 3.5. Lyophilization of mRNA-GFP

We used three cryoprotectants to stabilize and store naked mRNA after freeze-drying. We initially assessed mRNA integrity after lyophilization using the mRNA-GFP model. mRNA synthesis was performed as described in Section 2.3. Drying was performed as described in Section 2.7. The first cryoprotectant was sucrose-based, the second was mannose-based, and the third was trehalose-based. Samples were stored at 4 °C for one month. The lyophilized mRNA was then dissolved in nuclease-free water and used to transfect HEK293 cells. The results showed that after one month of storage, all preparations demonstrated high efficiency in GFP protein synthesis (Figure 4).

Based on the obtained results, we selected trehalose-based cryoprotectant No. 3 for further work. New mRNA-GFP preparations were prepared, lyophilized, and stored at −20 °C, +4 °C, and +20 °C for long periods. After 1 month and 3 months, lyophilized mRNA-GFP samples were dissolved in pure water and used to transfect HEK293 cells. The results showed that after both 1 month and 3 months of storage, all mRNA samples produced levels of GFP protein synthesis comparable to synthesized immediately before use prepared control mRNA; the differences were not statistically significant (Figure 4b,c). We also analyzed lyophilized mRNA using capillary electrophoresis. The results show that mRNA integrity after drying remains virtually unchanged compared to the control, synthesized immediately before use mRNA. All samples exhibited minor mRNA degradation, less than 5%, which is acceptable (Appendix A).

### 3.6. Immunogenicity Assessment of the Lyophilized Trivalent mRNA-Vaccine mRNA-Vector-Flu

The next stage of the study was to evaluate the immunogenic properties of the lyophilized trivalent seasonal influenza vaccine mRNA-Vector-Flu. The mRNA was lyophilized with trehalose-based cryoprotectant No. 3 and stored at +4 °C for one month.

To assess the immunogenicity of the mRNA-Vector-Flu vaccine, BALB\c mice (*n* = 6) were immunized twice with a three-week interval (Figure 5a). As a control, mice were immunized with synthesized immediately before use mRNA vaccine. Immunization was performed using the jet injection method.

Animal sera were collected on day 35 of the experiment. For ELISA, individual recombinant hemagglutinin proteins H1, H3, and HB, as well as their combination, were used as antigens. The results showed that both the synthesized immediately before use prepared and lyophilized mRNA seasonal influenza vaccine mRNA-Vector-Flu produced similar results.

The average titer for the recombinant H1 protein was 1:430,000 for the lyophilized vaccine and 1:870,000 for the synthesized immediately before use mRNA, 1:760,000 for H3, and 1:300,000 for HB. When all three recombinant hemagglutinin proteins were adsorbed in the ELISA, the average titer was 1:1,300,000 for both vaccines (Figure 5b).

To evaluate the ability of post-immunization sera to inhibit hemagglutination, a HAI assay was performed using three influenza viruses (Figure 5c). For A/Buryatia/106-6V/2022 (H1N1), both vaccine formulations produced similar hemagglutination inhibition titers, with an average titer of 1:40. Against A/Darwin/9/2021 (H3N2), the average titer was 1:40 for the lyophilized vaccine and 1:60 for synthesized immediately before use mRNA. For the B/Austria/1359417/2021 antigen, both vaccine variants elicited a higher response, with an average titer of 1:80.

An important criterion of vaccine effectiveness is its ability to induce antibodies capable of neutralizing the virus. Sera from mice immunized with both the lyophilized and non-lyophilized variant of the mRNA-Vector-FLU vaccine demonstrated the ability to neutralize influenza viruses A/Buryatia/106-6V/2022 (H1N1), A/Darwin/9/2021 (H3N2), and B/Austria/1359417/2021 in MDCK cell culture in vitro in the virus neutralization assay (Figure 5d). The average titer was approximately 1:100, except for the lyophilized vaccine with the A/Buryatia/106-6V/2022 (H1N1) virus, where the neutralization titer was 1:50.

For complete protection against viral infections, the development of a T-cell immune response is also necessary. This was assessed using IFN-γ-ELISpot. The results showed that, two weeks after the second immunization, T-cell immunity developed in mice immunized with both lyophilized and synthesized immediately before use mRNA vaccines in response to stimulation with a pool of specific peptides (Figure 5e). The average number of IFNγ-secreting T lymphocytes was 458 per 1 million cells in the group of animals immunized with lyophilized mRNA and 515 per 1 million cells in the group with synthesized immediately before use mRNA.

## 4. Discussion

mRNA vaccines offer a promising platform for developing new, effective influenza vaccines. Delivery of a naked mRNA vaccine using a needle-free jet injection system further enhances safety, reduces cost, and eliminates the need for lipid nanoparticles, which are traditionally used for mRNA delivery.

In the first phase of our work, we designed and produced a trivalent mRNA influenza vaccine, named mRNA-Vector-Flu, encoding the hemagglutinin (HA) of the seasonal influenza virus strains recommended by the WHO for the 2023–2024 season: A/Wisconsin/67/2022(H1N1)pdm09, A/Darwin/9/2021(H3N2), and B/Austria/1359417/2021. A study of the immunogenicity of the resulting constructs administered to mice using jet injection demonstrated that both the individual and trivalent mRNA-Vector-Flu mRNA influenza vaccines elicited a pronounced specific immune response in BALB/c mice (Figure 2). When immunized with the individual (monovalent) vaccines, each mRNA vaccine induced high titers of specific antibodies (mRNA-H1—1:400950, mRNA-H3—1:510300, mRNA-HB—1:225000), as demonstrated by ELISA using recombinant hemagglutinins corresponding to the specific vaccine as antigens. This confirms their ability to induce a strong and specific humoral response.

The control group immunized with the commercial Flu-M vaccine demonstrated significantly lower antibody titers, particularly to hemagglutinin H3, where the titer was virtually zero. This is due to the difference in the composition of the seasonal influenza strains in the vaccines, as the mRNA-Vector-Flu vaccine developed in this study has a composition for the 2023–2024 season, while the Flu-M vaccine used for the control group has a composition for the 2025–2026 season. Only the H3 component changed in the vaccine compositions for these years: in the 2023 season it was A/Darvin/9/2021 (H3N2), and in the 2025 season it was A/District of Columbia/27/2023 (H3N2).

In this study, we used a 10 μg dose of each immunogen for the mRNA-Vector-Flu-LNP vaccine, while we used a 30 μg dose of each immunogen for mRNA-Vector-Flu delivered using JI. We previously demonstrated that animals immunized with a 30 μg dose of mRNA-H5-LNP exhibited signs of stress, including fur ruffles, as well as increased discomfort during the second immunization session [34]. Similar results were reported by other authors studying different doses of the COVID-19 mRNA vaccine [37].

According to our results, trivalent mRNA encapsulated in LNPs induced slightly higher antibody levels compared to jet injection; however, the differences were not statistically significant. This suggests that jet injection is a promising mRNA delivery method, comparable in efficacy to traditional LNPs.

Thus, the obtained results show that the developed mRNA vaccine mRNA-Vector-Flu elicits a strong specific immune response, and the components of the trivalent vaccine are able to maintain their immunogenicity in combination without suppressing each other’s immune responses.

Analysis of the dose-dependent immune response of the vaccine revealed varying effects of different doses on immune response formation (Figure 3). According to the obtained results, at a vaccine dose of 150 µg (50 µg of each immunogen), antibody titers were approximately equal to those observed at a 30 µg dose (10 µg of each immunogen). It is possible that high doses of mRNA molecules may lead to immunosuppression. A dose of 90 µg (30 µg of each immunogen) showed good results. Thus, the optimal dose for an mRNA vaccine against seasonal influenza is 10–30 µg of each immunogen.

It is known that in aqueous solutions, naked mRNA preparations are relatively unstable; one possible approach to stabilize the formulation is lyophilization. Therefore, the next stage of the study was to determine the feasibility of lyophilization and storage of naked mRNA.

We conducted preliminary studies on the effect of lyophilization on the biological properties of the formulation using an mRNA-GFP model. mRNA-GFP samples lyophilized with three different cryoprotectants, selected based on literature data [38,39], demonstrated comparable efficiency of GFP protein synthesis in transfected cells after one month of storage at +4 °C (Figure 4a). Based on these results, a trehalose-based cryoprotectant was selected for further work. mRNA-GFP lyophilized using trehalose and stored at –20 °C, +4 °C, and +20 °C for three months maintained GFP protein synthesis at a level comparable to synthesized immediately before use control mRNA (Figure 4b,c). The stability of lyophilized mRNA at positive temperatures may, in the future, help address the challenges of storage and transportation of mRNA vaccines.

The third stage of the study focused on evaluating the biological properties of the trivalent mRNA vaccine against seasonal influenza, mRNA-Vector-Flu, lyophilized with trehalose and stored at +4 °C for one month. A synthesized immediately before use mRNA vaccine was used as a control. Immunization with the trivalent mRNA-Vector-Flu, containing 15 µg of each component in 50 µL of saline, was performed via jet injection, and animal sera were collected on day 35 of the experiment.

To quantify the humoral response, an ELISA method was used, using both individual recombinant hemagglutinin proteins H1, H3, and HB, as well as a combination of them, as antigens. The data obtained demonstrated that the lyophilized mRNA vaccine retains a high level of immunogenicity comparable to that of the synthesized immediately before use mRNA vaccine (Figure 5b). The average antibody titer against the H1 protein was 1:430,000 for the lyophilized vaccine and 1:870,000 for synthesized immediately before use mRNA. For H3, it reached 1:760,000, and for HB, 1:300,000. When all three proteins were simultaneously adsorbed in an ELISA, the average titer for both vaccines was 1:1,300,000, indicating the formation of a strong specific humoral response against different strains of the seasonal influenza virus. The HAI and microneutralization assay results demonstrated that both the lyophilized and synthesized immediately before use mRNA-Vector-Flu vaccines were capable of neutralizing all three seasonal influenza viruses (Figure 5c,d). In addition to humoral immunity, the eliciting of a T-cell response is important for comprehensive protection against viral infection. This was assessed using an IFN-γ ELISpot assay, using a pool of specific peptides to stimulate cells. The results showed that two weeks after the second immunization, animals receiving both lyophilized and synthesized immediately before use mRNA vaccines developed a robust T-cell immune response. The average number of IFN-γ-secreting T cells was 458 per million cells in the lyophilized mRNA group and 515 per million cells in group of animals immunized with synthesized immediately before use the mRNA (Figure 5e).

Thus, the lyophilized mRNA-Vector-Flu mRNA vaccine, stored at 4 °C for a month, demonstrates the ability to induce both a strong humoral and cellular immune response comparable to the effect of the synthesized immediately before use prepared mRNA vaccine. These data confirm the potential of the lyophilized vaccine for further use and extended shelf life without loss of immunogenic activity.

The next step in our research is to test the biological activity of lyophilized naked mRNA-Vector-Flu stored at 4 °C and 20 °C for a year.

Combining lyophilization of the naked mRNA vaccine with our developed method of delivering mRNA via jet injection solves the main problem of prophylactic mRNA vaccine platforms: delivery. Using the gold standard of delivery, lipid nanoparticles (LNPs), significantly increases the cost of vaccine production; LNPs can cause undesirable side effects; mRNA-LNPs must be stored at extremely low temperatures or a complex cryoprotectant composition must be used for their lyophilization [40].

## 5. Conclusions

In this study, we produced and characterized a trivalent mRNA vaccine against the seasonal influenza virus, mRNA-Vector-Flu. Delivery of the naked vaccine via jet injection was shown to elicit a high immune response in BALB/c mice. Lyophilization with a trehalose-based cryoprotectant was used for stabilization and storage. Using the mRNA-GFP model, we demonstrated that lyophilized mRNA remained stable during storage for at least 3 months at +4 °C and +20 °C. Notably, the biological activity of lyophilized mRNA-GFP was comparable to that of synthesized immediately before use prepared mRNA. A comparative study of the immunogenicity of naked mRNA-Vector-Flu showed that both the synthesized immediately before use the mRNA preparation and the lyophilized form stored at +4 °C for a month induced similar levels of virus-specific antibodies and generated a pronounced T-cell immune response.

Thus, delivery of a naked mRNA vaccine using a needle-free jet injection ensures a high-level immune response, which improves its safety, reduces its cost, and eliminates the need for lipid nanoparticles traditionally used for mRNA delivery. At the same time, lyophilization of the naked mRNA vaccine preserves its biological activity and ensures its storage for at least a month at +4 °C temperatures. Our results demonstrate that our proposed approach can be considered a promising direction for the development and improvement of the mRNA vaccine platform.

## Figures and Tables

**Figure 1 vaccines-14-00056-f001:**
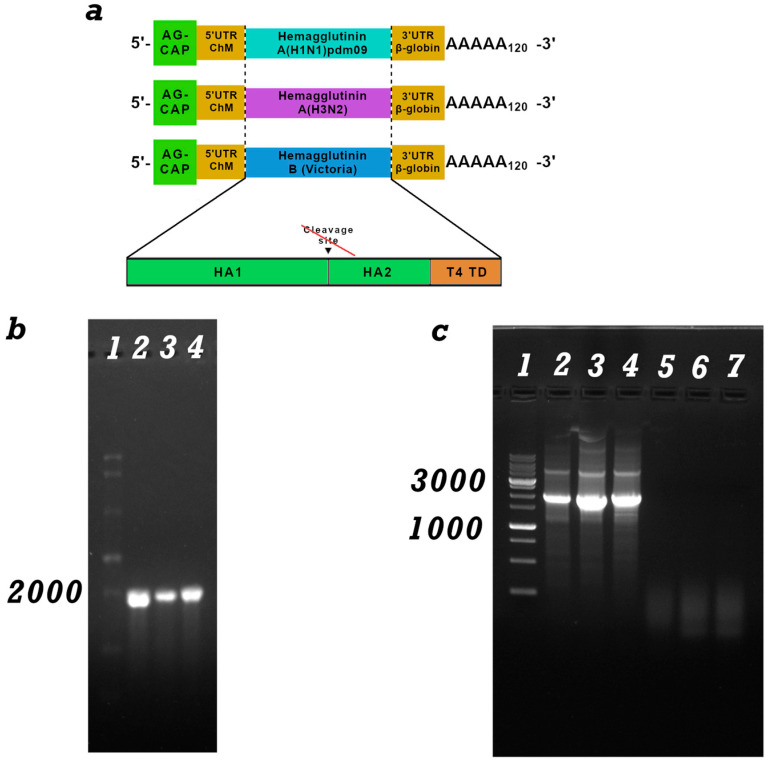
(**a**) Schematic representation of mRNA: mRNA has a 5′-cap, poly(A) tail, 5′- and 3′-untranslated regions of ChM and human β-globin, an open reading frame encoding hemagglutinin (HA) of influenza H1N1, H3N2 and Victoria B; Schematic structure of HA genes modified by the trimerizing domain of phage T4 (T4 TD); (**b**) Electropherogram of mRNA synthesis products in 1% agarose gel: lane 1-ssRNA Ladder (New England Biolabs); 2-mRNA-H1, 3-mRNA -H3 and 3-mRNA-HB; (**c**) Electropherogram of PCR products from the mRNA preparation for the presence of a DNA template. lanes 1-DNA marker, 2-linearized DNA template pVAX-C3-H1-24, 2-linearized DNA template pVAX-C3-H3-24, 3-linearized DNA template pVAX-C3-HB-24, 4-mRNA-H1, 5-mRNA-H3 and 6-mRNA-HB.

**Figure 2 vaccines-14-00056-f002:**
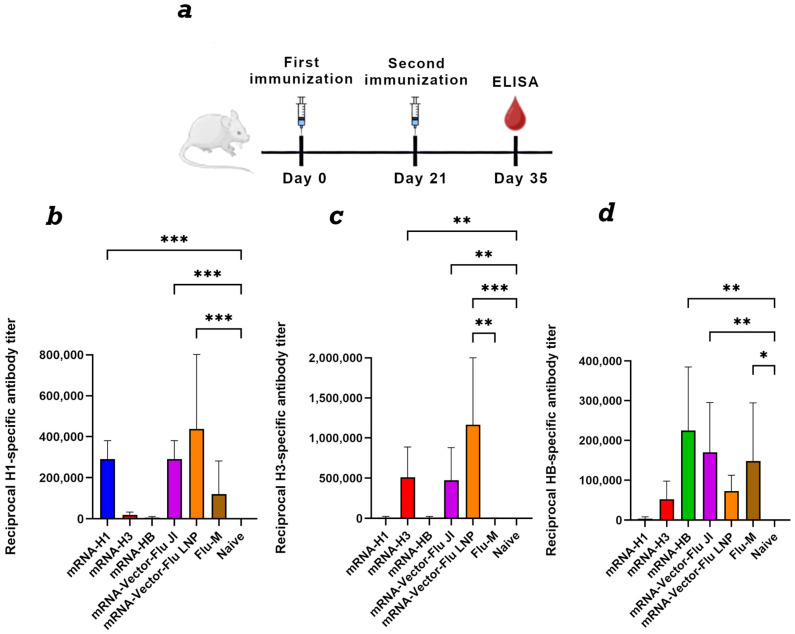
Study of the humoral immune response to components of the trivalent mRNA vaccine against the seasonal influenza virus. (**a**) Immunization schedule. (**b**) H1-specific antibody titer (ELISA data) (number of animals: *n* = 6). (**c**) H3-specific antibody titer (ELISA data) (number of animals: *n* = 6). (**d**) HB-specific antibody titer (ELISA data) (number of animals: *n* = 6). Reciprocal titer values are provided in the plots. mRNA-H1—a group of animals immunized with mRNA-H1; mRNA-H3—a group of animals immunized with mRNA-H3; mRNA-HB—a group of animals immunized with mRNA-HB; mRNA-Vector-Flu JI—a group of animals immunized with a trivalent mRNA vaccine delivered by JI; mRNA-Vector-Flu LNP—a group of animals immunized with a trivalent mRNA vaccine delivered by LNP; Flu-M—a group of animals immunized with the commercial vaccine Flu-M; Naive—a group of non-immunized animals. In panels, data are provided as means with full range. * *p* < 0.05, ** *p* < 0.01, *** *p* < 0.001 following Kruskal–Wallis analysis of variance with correction for multiple comparisons and Dunn’s statistical hypothesis test.

**Figure 3 vaccines-14-00056-f003:**
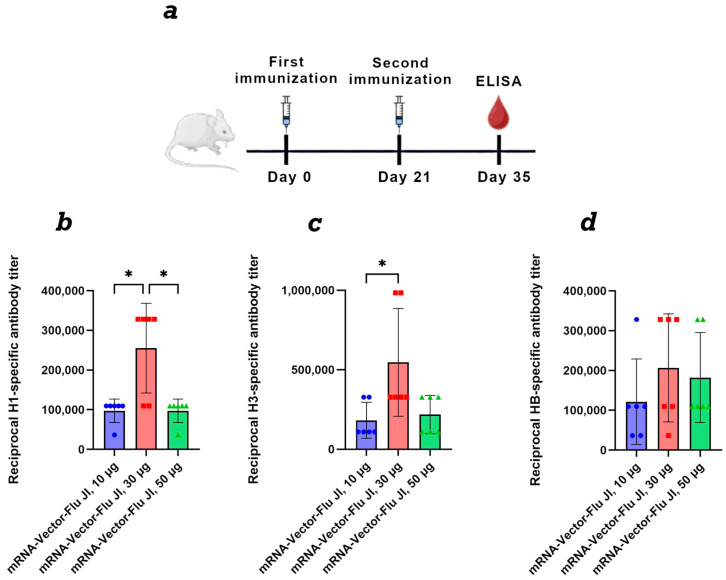
Study of the humoral immune response of a trivalent mRNA vaccine against the seasonal influenza virus depending on the dose. (**a**) Immunization schedule. (**b**) H1-specific antibody titer (ELISA data) (number of animals: *n* = 6). (**c**) H3-specific antibody titer (ELISA data) (number of animals: *n* = 6). (**d**) HB-specific antibody titer (ELISA data) (number of animals: *n* = 6). Reciprocal titer values are provided in the plots. mRNA-Vector-Flu JI, 10 μg—a group of animals immunized with a trivalent mRNA vaccine, the dose of each component was 10 μg, a total of 30 μg; mRNA-Vector-Flu JI, 30 μg—a group of animals immunized with a trivalent mRNA vaccine, the dose of each component was 30 μg, a total of 90 μg; mRNA-Vector-Flu JI, 50 μg—a group of animals immunized with a trivalent mRNA vaccine, the dose of each component was 50 μg, a total of 150 μg. In panels, data are provided as means with full range. * *p* < 0.05 following Kruskal–Wallis analysis of variance with correction for multiple comparisons and Dunn’s statistical hypothesis test.

**Figure 4 vaccines-14-00056-f004:**
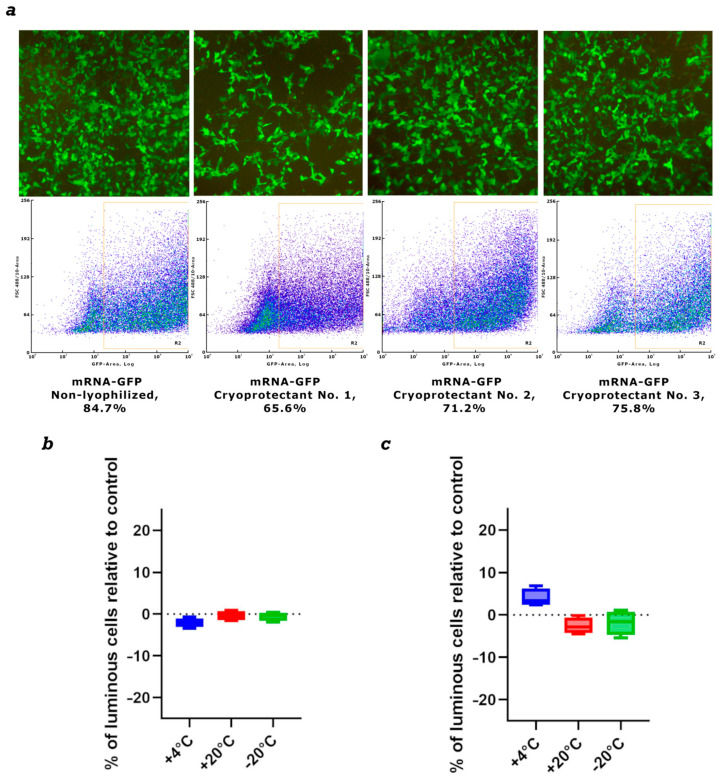
Evaluation of mRNA stability after lyophilization. (**a**) Synthesis of GFP protein in HEK293 cells transfected with mRNA-GFP after lyophilization in different cryoprotectants. (**b**) The level of GFP protein synthesis in HEK293 cells transfected with lyophilized mRNA-GFP after storage for one month. (**c**) The level of GFP protein synthesis in HEK293 cells transfected with lyophilized mRNA-GFP after storage for three months. Data are shown as a comparison with the control synthesized immediately before use mRNA-GFP, where the control is 0. +4 °C—lyophilized mRNA-GFP stored at +4 °C; +20 °C—lyophilized mRNA-GFP stored at +20 °C; −20 °C—lyophilized mRNA-GFP stored at −20 °C.

**Figure 5 vaccines-14-00056-f005:**
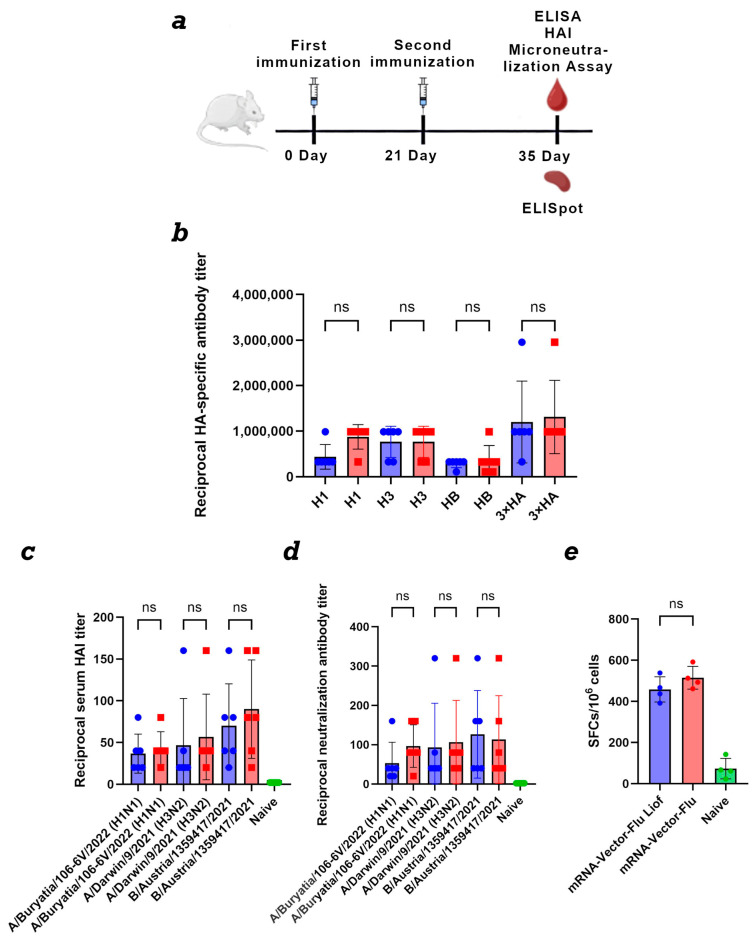
Comparison of the immunogenicity of lyophilized and synthesized immediately before use mRNA vaccine mRNA-Vector-Flu. (**a**) Immunization schedule. (**b**) Study of the humoral immune response of the lyophilized mRNA vaccine mRNA-Vector-Flu. H1—the recombinant protein H1 was used as the antigen in the ELISA; H3—the recombinant protein H3 was used as the antigen in the ELISA; HB—the recombinant protein HB was used as the antigen in the ELISA; 3 × HA—a mixture of recombinant HA proteins was used as the antigen in the ELISA. Lyophilized mRNA-Vector-Flu is shown in blue, synthesized immediately before use mRNA-Vector-Flu is shown in red. (**c**) Results of hemagglutination-inhibition reaction with A/Buryatia/106-6V/2022 (H1N1), A/Darwin/9/2021 (H3N2), and B/Austria/1359417/2021. Vector-Flu is shown in blue, synthesized immediately before use mRNA-Vector-Flu is shown in red. (**d**) Microneutralization antibody titers against influenza virus A/Buryatia/106-6V/2022 (H1N1), A/Darwin/9/2021 (H3N2), and B/Austria/1359417/2021. Vector-Flu is shown in blue, synthesized immediately before use mRNA-Vector-Flu is shown in red. (**e**) ELISpot assay results of specific T-cell responses in immunized BALB/c mice. Number of cells expressing IFN-γ in response to stimulation with a pool of HA-specific peptides per 1 × 106 splenocytes. Data are presented as the median with a range of inverse titers. Significance was assessed using non-parametric one-factor Kruskal–Wallis analysis of variance (ns—not statistically significant). mRNA-Vector-Flu Liof—mice immunized with lyophilized mRNA; mRNA-Vector-Flu—mice immunized with synthesized immediately before use mRNA.

## Data Availability

The data can be shared upon request.

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
