# Peer review of "A Naked Lyophilized mRNA Vaccine Against Seasonal Influenza, Administered by Jet Injection, Provides a Robust Response in Immunized Mice"

_vaccines, 2026, doi:10.3390/vaccines14010056_

Round 1

Reviewer 1 Report

Comments and Suggestions for Authors

This manuscript described a series of experiments on using “jet injection” immunization to evaluate the “stability” and immunogenicity of a mRNA encoding the hemagglutinin (of influenza virus) in vivo.

Presumably, the study described in this manuscript is part of this research group’s effort in developing jet injection for “naked mRNA” as a vaccination platform, to circumvent the adverse effects of liposome-encapsulated mRNA vaccines.  

There had been comprehensive experiments and data in the literature to indicate the feasibility of this “jet injection” vaccination platform. However, in this manuscript, there are many deficiencies that need to be addressed prior to publication:

  1. The authors stated that the aim was to determine the stability and integrity of mRNA after lyophilization. This aim could be achieved simply by physical methods or by RNA sequencing, i.e., by in vitro methods (e.g., Fig. 4). Conducting a comprehensive immunogenicity study seems to be excessive and unnecessary.
  2. If the aim was to evaluate jet injection for a trivalent influenza vaccine, it should be stated so. And if there are other manuscripts in parallel, it should be disclosed.
  3. Although it referred to reference #27, the exact parameters (of jet injection) should be described, e.g., spring-load jet, energy level, and duration.
  4. Current understanding of the cause of adverse reactions by LNP-mRNA vaccine is due to its prolonged and unknown dissemination in vivo, and by the modified nucleotides. Whereas jet injections may circumvent the adverse effect of LNP, it is not known why they still use modified U in the mRNA (Line 127). This modification may also prolong the half-life of jet injected mRNA and may cause similar adverse effects. Therefore, the authors need to examine this.
  5. The assays for the immune responses were not according to standardized protocols or parameters. For example, the unit for ELISA is unknown (and at high value!). Other standardized assays for influenza immunization should be used, e.g., Hemagglutination-inhibition (according to WHO protocol), or cell-based neutralization assay. Likewise, the isotype of antibodies should be identified.
  6. Minor corrections/issues: Use of Gmail address instead of official email address; Line 491: eliciting T-cell response, not “formation”, etc. And there are some repetitive sentences that need to be edited. 
Comments on the Quality of English Language

NA

Author Response

Reviewer 1:

This manuscript described a series of experiments on using “jet injection” immunization to evaluate the “stability” and immunogenicity of a mRNA encoding the hemagglutinin (of influenza virus) in vivo.

Presumably, the study described in this manuscript is part of this research group’s effort in developing jet injection for “naked mRNA” as a vaccination platform, to circumvent the adverse effects of liposome-encapsulated mRNA vaccines.  

There had been comprehensive experiments and data in the literature to indicate the feasibility of this “jet injection” vaccination platform. However, in this manuscript, there are many deficiencies that need to be addressed prior to publication:

Comments 1: The authors stated that the aim was to determine the stability and integrity of mRNA after lyophilization. This aim could be achieved simply by physical methods or by RNA sequencing, i.e., by in vitro methods (e.g., Fig. 4). Conducting a comprehensive immunogenicity study seems to be excessive and unnecessary.

Response 1: Thank you for your comment. In our future work, we plan to conduct preclinical studies of the mRNA-Vector-Flu vaccine. Therefore, it was important for us to demonstrate its immunogenicity after lyophilization and storage. Accordingly, we have presented data on its stability both in vitro and in vivo.

Comments 2: If the aim was to evaluate jet injection for a trivalent influenza vaccine, it should be stated so. And if there are other manuscripts in parallel, it should be disclosed.

Response 2: Indeed, one of the objectives of our study was to demonstrate the jet injection method for mRNA vaccine delivery, not limited to influenza vaccines. Our research group was among the first in the world to propose this delivery approach. Below is a list of our key publications on this topic:

  1. Kisakov D.N., Kisakova L.A., Sharabrin S.V., Yakovlev V.A., Tigeeva E.V., Borgoyakova M.B., et al. Delivery of Experi-mental MRNA Vaccine, Encoding the RBD of SARS-CoV-2 Using Jet Injection. Bull Exp Biol Med. 2023;176(6):751-6
  2. Kisakov D.N., Karpenko L.I., Kisakova L.A., Sharabrin S.V., Borgoyakova M.B., Starostina E.V., et al. Jet Injection of Naked mRNA Encoding the RBD of the SARS-CoV-2 Spike Protein Induces a High Level of a Specific Immune Response in Mice. Vaccines. 2025;13(1):65
  3. Sharabrin, S.V., Ilyichev, A.A., Kisakov, D.N. et al. Needle-Free Jet-Delivered mRNA-Vaccine Encoding Influenza A(H1N1)pdm09 Hemagglutinin Protects Mice from Lethal Virus Infection. Mol Biol 59, 376–389 (2025)
  4. Yakovlev V.A., Litvinova V.R., Rudometova N.B., et al. Immunogenic and Protective Properties of mRNA Vaccine Encod-ing Hemagglutinin of Avian Influenza A/H5N8 Virus, Delivered by Lipid Nanoparticles and Needle-Free Jet Injection. Vaccines 2025;13(8):883

Comments 3: Although it referred to reference #27, the exact parameters (of jet injection) should be described, e.g., spring-load jet, energy level, and duration.

Response 3: Thank you for your comment. We have added this information to Section 2.8.

Comments 4: Current understanding of the cause of adverse reactions by LNP-mRNA vaccine is due to its prolonged and unknown dissemination in vivo, and by the modified nucleotides. Whereas jet injections may circumvent the adverse effect of LNP, it is not known why they still use modified U in the mRNA (Line 127). This modification may also prolong the half-life of jet injected mRNA and may cause similar adverse effects. Therefore, the authors need to examine this.

Response 4: Thank you for your comment. We believe that the adverse effects observed with mRNA-LNP formulations are primarily caused by the lipids rather than the modified mRNA itself. CureVac conducted clinical trials of its COVID-19 vaccine without using pseudouridine, instead optimizing the mRNA sequence, including uridine depletion. However, their results showed a vaccine efficacy of approximately 48%.

Regarding our studies, we have investigated the impact of modified nucleotides both in vitro using GFP and in vivo with an influenza mRNA vaccine delivered via jet injection. These results are currently in press:

Krasnikova S.I., Sharabrin S.V., Kisakov D.N., Borgoyakova M.B., Starostina E.V., Tigeeva E.V., Makarova K.P., Yakovlev V.A., Ivanova K.I., Rudometov A.P., Ilyichev A.A., Karpenko L.I. Effect of modified nucleotides on the immunogenicity of mRNA vaccine against H1N1 influenza // Bull. Exp. Biol. 2025. Vol. 180, No. 11. P. 000–000. EDN: ZSMBSW; doi: 10.47056/0365-9615-2025-180-11-

Our findings indicated that the mRNA vaccine containing unmodified uridine, when delivered by jet injection, elicited lower humoral and cellular immune responses compared to the pseudouridine-containing formulation.

Comments 5: The assays for the immune responses were not according to standardized protocols or parameters. For example, the unit for ELISA is unknown (and at high value!). Other standardized assays for influenza immunization should be used, e.g., Hemagglutination-inhibition (according to WHO protocol), or cell-based neutralization assay. Likewise, the isotype of antibodies should be identified.

Response 5: We thank the reviewer for this constructive comment. Indeed, for influenza vaccines, virus neutralization is an important criterion of efficacy. At the time the experiments were conducted, scheduled maintenance was being carried out in our BSL-2 laboratories, which prevented work with live virus. Therefore, the manuscript was initially submitted without these data. However, serum samples from the experimental mice had been preserved, and we have now performed the neutralization assays. The results have been added to the revised manuscript.

Comments 6: Minor corrections/issues: Use of Gmail address instead of official email address; Line 491: eliciting T-cell response, not “formation”, etc. And there are some repetitive sentences that need to be edited.

Response 6: The personal email address is provided due to the inaccessibility of the work email outside of office hours. The personal address ensures timely receipt of correspondence from the journal.

Reviewer 2 Report

Comments and Suggestions for Authors

The authors reported a lyophilised mRNA vaccine against seasonal influenza, administered by jet injection, and claimed that it provided a robust response in immunised animals. After a careful reading, this reviewer finds that the manuscript is overall well written. The study design is appropriate for testing the study question. Methods used are robust. The results and discussion are comprehensive. I would suggest a few minor changes, as follows:

In the title of the manuscript, please mention that mice were used instead of 'animals', as this can be confusing for the reader.  Also, I suggest that English could be improved in a few places for better presentation. Other specific comments are the following: 

Line 47: Please replace "annual re-release" with a more appropriate term.

Lines 71-80: Merge both paragraphs. 

Lines 81-82: A paragraph of just one sentence is not appropriate.

Line 88: Neb should be written as NEB, which stands for New England Biolabs. 

Line 89: Please explain how DNA templates were obtained and cloned for IVT.

Lines 94-95: Please explain these three plasmids in detail.

Line 119: Please provide the catalogue number of the HiPure Plasmid Mini Kit used, as well as any other kits used in the methodology. 

Comments on the Quality of English Language

English could be improved. 

Author Response

Reviewer 2:

The authors reported a lyophilised mRNA vaccine against seasonal influenza, administered by jet injection, and claimed that it provided a robust response in immunised animals. After a careful reading, this reviewer finds that the manuscript is overall well written. The study design is appropriate for testing the study question. Methods used are robust. The results and discussion are comprehensive. I would suggest a few minor changes, as follows:

In the title of the manuscript, please mention that mice were used instead of 'animals', as this can be confusing for the reader.  Also, I suggest that English could be improved in a few places for better presentation. Other specific comments are the following: 

Comments 1: Line 47: Please replace "annual re-release" with a more appropriate term.

Lines 81-82: A paragraph of just one sentence is not appropriate.

Line 88: Neb should be written as NEB, which stands for New England Biolabs.

Response 1: We thank you for your comments and have made the suggested revisions accordingly.

Comments 2: Line 89: Please explain how DNA templates were obtained and cloned for IVT.

Response 2: We thank the reviewer for this comment. Indeed, this part was insufficiently described. We have revised the paragraph and provided a more detailed description of the preparation of DNA templates in Section 2.2.

Comments 3: Lines 94-95: Please explain these three plasmids in detail.

Response 3: We have supplemented the manuscript with a description of the plasmids.

Comments 4: Line 119: Please provide the catalogue number of the HiPure Plasmid Mini Kit used, as well as any other kits used in the methodology

Response 4: We have supplemented the manuscript by adding catalog numbers for several of the kits used.

Reviewer 3 Report

Comments and Suggestions for Authors

Summary

This manuscript describes the design and preclinical evaluation of a trivalent mRNA vaccine encoding influenza virus antigen, HA. The vaccine is delivered as naked mRNA via needle-free jet injection, and the authors compare based on formulations, delivery methods, and storage condition. They show high ELISA titers against recombinant HA and IFNG ELISpot responses in BALB/c mice, and they demonstrate stability of lyophilized mRNA up to 3 months under various storage conditions. The overall findings and topic are interesting and timely. The combination of naked mRNA, jet injection, and lyophilization for near-ambient storage is potentially interesting. However, several aspects of the work and its presentation require substantial strengthening and revision.

Major comments

1. The central conclusion is that mRNA-Vector-Flu “provides a robust response” and is a promising platform for seasonal influenza vaccination. However, the immunogenicity assessment relies entirely on ELISA against recombinant HA antigens and IFN-γ ELISpot using HA-derived peptides.

For influenza vaccines, hemagglutination inhibition (HAI) and/or virus neutralization assays are standard correlates of protection and functional evaluation. Without these data, it is difficult to judge whether the antibody responses are functionally protective. So, I recommend HAI and/or neutralization assay (ideally with real virus, but if it not feasible, with pseudoviral assay).

If challenge experiments are not possible at this stage, this limitation should be clearly acknowledged in the Discussion and Conclusions, and any implication of proven protection should be toned down.

2. The abstract and conclusions state that lyophilization “ensures its long-term storage at above-zero temperatures (line 32)” and “long-term storage at elevated temperatures. (line 528)” However, for the vaccine itself, the lyophilized trivalent mRNA-Vector-Flu was stored at +4 °C for 1 month before testing. The longer-term (3-month) stability data are only shown for the mRNA-GFP model, not for the actual vaccine construct. Since RNA sequences and secondary structure also could affect stability during storage conditions, it would be better to substantially soften the wording if the authors cannot provide additional stability data for mRNA-Vector-Flu (ex. 3 months at +4 and +20 °C).

3. There are several inconsistencies that the authors need to clarify. Methods state that each group typically contains 6 mice. However, in figure 2 legend reports n=16 for H1, but n=6 for H3 and HB. Also, some sections mention sera collected on day 28, but Immunization schedule figures say day 35. Please provide a clear table or description for each experiment (group size, dose, route, injection device, time of immunizations, and time of sample collection). And please clarify why n differs between antigens in Figure 2 (were samples missing, or just typo?), and ensure that time points in the text, figures, and legends are fully consistent.

4. The comparison between jet injection and LNP delivery is a key point of the manuscript, but some details are missing and need to be clarified further.

For jet injection, please provide more details for the injection method. Specify device manufacturer/model, injection pressure or settings, needle-free nozzle type, injection depth, and exact injection volume per shot. This information is not provided from cited paper (#34).

The manuscript uses 10 µg per component for LNP-mRNA immunization and 30 µg per component for naked mRNA delivered by jet injection. However, the authors do not provide a clear rationale for this 3-fold difference in dose. Because mRNA dose directly influences immunogenicity, this discrepancy complicates interpretation of the comparative performance of LNP vs jet-injected mRNA. For example, in one of the author-cited reference paper (#27), they were using same amount of mRNA (30 µg) for comparing JI and LNP. Please clarify how these doses were selected (whether based on previous optimization, literature precedent, or expected differences in bioavailability) and justify why these doses constitute an appropriate comparison.

5. The reference #33, I cannot find the paper having title “Development of a Needle-Free Method for Delivery of mRNA Vaccines Using a Jet Injector”. Since this reference includes several important information, such as original methods and rational for this manuscript, please double check the reference information and provide revised information for further reviewing.

Minor comments

1. Please review overall paper and revise all typos and/or errors. Below are some of the examples but not limited to;

  • 2 - mRNA-H1, 3 - mRNA -H3 and 3 - mRNA-H3 (line 297, in Figure 1 lagend)
  • Many “μg”s are written as “ug” in manuscript, figure and figure legend. Please revise them.
  • Several abbreviations are missing. (SFCs, HA, etc.). Please check overall manuscript and revise them.

2. In Section 2.7, lyophilization conditions are described as “lyophilized for 24 hours at 22±2 °C”. Since the authors described more detail condition (freezing samples at -50±2ºC for 8 hours, and lyophilizing samples at 22±2ºC for the remainder of the time., lines 171-172), maybe deleting “22±2 °C” from line 170 would make less confusion for the readers.

3. To make the results clearer, please consider plotting individual data points to show variability, especially given the relatively small group sizes.

Author Response

Reviewer 3:

This manuscript describes the design and preclinical evaluation of a trivalent mRNA vaccine encoding influenza virus antigen, HA. The vaccine is delivered as naked mRNA via needle-free jet injection, and the authors compare based on formulations, delivery methods, and storage condition. They show high ELISA titers against recombinant HA and IFNG ELISpot responses in BALB/c mice, and they demonstrate stability of lyophilized mRNA up to 3 months under various storage conditions. The overall findings and topic are interesting and timely. The combination of naked mRNA, jet injection, and lyophilization for near-ambient storage is potentially interesting. However, several aspects of the work and its presentation require substantial strengthening and revision.

Major comments

Comments 1: The central conclusion is that mRNA-Vector-Flu “provides a robust response” and is a promising platform for seasonal influenza vaccination. However, the immunogenicity assessment relies entirely on ELISA against recombinant HA antigens and IFN-γ ELISpot using HA-derived peptides.

For influenza vaccines, hemagglutination inhibition (HAI) and/or virus neutralization assays are standard correlates of protection and functional evaluation. Without these data, it is difficult to judge whether the antibody responses are functionally protective. So, I recommend HAI and/or neutralization assay (ideally with real virus, but if it not feasible, with pseudoviral assay).

If challenge experiments are not possible at this stage, this limitation should be clearly acknowledged in the Discussion and Conclusions, and any implication of proven protection should be toned down.

Response 1: We thank the reviewer for this constructive comment. Indeed, for influenza vaccines, the ability to neutralize the virus is a key criterion of efficacy. At the time the experiments were conducted, scheduled preventive maintenance was being carried out in our BSL-2 laboratories, which prevented us from working with live influenza virus. Therefore, the manuscript was initially submitted without these data. However, serum samples from the experimental mice had been preserved, and we have now performed the corresponding neutralization studies. The results of these experiments have been added to the revised manuscript.

Since mice are not a fully relevant animal model for seasonal influenza viruses (or require the use of mouse-adapted strains), challenge experiments were not conducted in this study. In future work, challenge studies are planned to be performed in ferrets.

Comments 2: The abstract and conclusions state that lyophilization “ensures its long-term storage at above-zero temperatures (line 32)” and “long-term storage at elevated temperatures. (line 528)” However, for the vaccine itself, the lyophilized trivalent mRNA-Vector-Flu was stored at +4 °C for 1 month before testing. The longer-term (3-month) stability data are only shown for the mRNA-GFP model, not for the actual vaccine construct. Since RNA sequences and secondary structure also could affect stability during storage conditions, it would be better to substantially soften the wording if the authors cannot provide additional stability data for mRNA-Vector-Flu (ex. 3 months at +4 and +20 °C).

Response 2: Thank you for your comment. Indeed, we did not present storage data for the lyophilized mRNA-Vector-Flu vaccine at +20 °C for periods longer than one month, as the formulation is currently undergoing long-term stability testing for up to one year. We have revised the wording in the manuscript accordingly.

Comments 3: There are several inconsistencies that the authors need to clarify. Methods state that each group typically contains 6 mice. However, in figure 2 legend reports n=16 for H1, but n=6 for H3 and HB. Also, some sections mention sera collected on day 28, but Immunization schedule figures say day 35. Please provide a clear table or description for each experiment (group size, dose, route, injection device, time of immunizations, and time of sample collection). And please clarify why n differs between antigens in Figure 2 (were samples missing, or just typo?), and ensure that time points in the text, figures, and legends are fully consistent.

Response 3: Indeed, the time points were incorrectly indicated in the figures. We have corrected them both in the figures and in the text. All experiments were conducted according to the same schedule, as described in the Materials and Methods (Section 2.8): the second immunization was performed on day 21, and the experiment was completed on day 35. The discrepancy in the figure legend was due to a typographical error; the correct number of animals is n = 6.

Comments 4: The comparison between jet injection and LNP delivery is a key point of the manuscript, but some details are missing and need to be clarified further.

For jet injection, please provide more details for the injection method. Specify device manufacturer/model, injection pressure or settings, needle-free nozzle type, injection depth, and exact injection volume per shot. This information is not provided from cited paper (#34).

The manuscript uses 10 µg per component for LNP-mRNA immunization and 30 µg per component for naked mRNA delivered by jet injection. However, the authors do not provide a clear rationale for this 3-fold difference in dose. Because mRNA dose directly influences immunogenicity, this discrepancy complicates interpretation of the comparative performance of LNP vs jet-injected mRNA. For example, in one of the author-cited reference paper (#27), they were using same amount of mRNA (30 µg) for comparing JI and LNP. Please clarify how these doses were selected (whether based on previous optimization, literature precedent, or expected differences in bioavailability) and justify why these doses constitute an appropriate comparison.

Response 4: We have added details on jet injection and updated the references in Section 2.8. Indeed, the dose of mRNA-LNP was three times lower than that of mRNA administered via jet injection (JI). In one of our previous studies, we directly compared the doses using both mRNA vaccine delivery methods. We observed that animals immunized with mRNA-H5-LNP exhibited signs of stress, including piloerection and increased discomfort during the second immunization procedure. Similar observations have been reported by other authors. This information has now been added to the Discussion section (Lines 502–511).

Comments 5: The reference #33, I cannot find the paper having title “Development of a Needle-Free Method for Delivery of mRNA Vaccines Using a Jet Injector”. Since this reference includes several important information, such as original methods and rational for this manuscript, please double check the reference information and provide revised information for further reviewing.

Response 5: We apologize for the incorrect reference provided earlier; it has now been corrected. Below is the list of our studies on the use of jet injection for mRNA vaccine delivery:

  1. Kisakov D.N., Kisakova L.A., Sharabrin S.V., Yakovlev V.A., Tigeeva E.V., Borgoyakova M.B., et al. Delivery of Experi-mental MRNA Vaccine, Encoding the RBD of SARS-CoV-2 Using Jet Injection. Bull Exp Biol Med. 2023;176(6):751-6
  2. Kisakov D.N., Karpenko L.I., Kisakova L.A., Sharabrin S.V., Borgoyakova M.B., Starostina E.V., et al. Jet Injection of Naked mRNA Encoding the RBD of the SARS-CoV-2 Spike Protein Induces a High Level of a Specific Immune Response in Mice. Vaccines. 2025;13(1):65
  3. Sharabrin, S.V., Ilyichev, A.A., Kisakov, D.N. et al. Needle-Free Jet-Delivered mRNA-Vaccine Encoding Influenza A(H1N1)pdm09 Hemagglutinin Protects Mice from Lethal Virus Infection. Mol Biol 59, 376–389 (2025)
  4. Yakovlev V.A., Litvinova V.R., Rudometova N.B., et al. Immunogenic and Protective Properties of mRNA Vaccine Encod-ing Hemagglutinin of Avian Influenza A/H5N8 Virus, Delivered by Lipid Nanoparticles and Needle-Free Jet Injection. Vaccines 2025;13(8):883

Comments 6: Minor comments

  1. Please review overall paper and revise all typos and/or errors. Below are some of the examples but not limited to;
  • 2 - mRNA-H1, 3 - mRNA -H3 and 3 - mRNA-H3 (line 297, in Figure 1 lagend)
  • Many “μg”s are written as “ug” in manuscript, figure and figure legend. Please revise them.
  • Several abbreviations are missing. (SFCs, HA, etc.). Please check overall manuscript and revise them.
  1. In Section 2.7, lyophilization conditions are described as “lyophilized for 24 hours at 22±2 °C”. Since the authors described more detail condition (freezing samples at -50±2ºC for 8 hours, and lyophilizing samples at 22±2ºC for the remainder of the time., lines 171-172), maybe deleting “22±2 °C” from line 170 would make less confusion for the readers.
  2. To make the results clearer, please consider plotting individual data points to show variability, especially given the relatively small group sizes.

Response 6: We appreciate your comments and have made the corresponding revisions.

Round 2

Reviewer 2 Report

Comments and Suggestions for Authors

Thanks for considering my previous suggestions and improving the manuscript. 

Author Response

Dear reviewer, thank you very much for your work; you have helped to make our manuscript better.

Reviewer 3 Report

Comments and Suggestions for Authors

Thank you for revising the manuscript. Most of the concerns raised have been addressed. However, there are a few minor things that were either not revised or not explained.

1. In Figure 3, “μg” is still written as “ug.” Please correct the unit notation.

2. Similar to Figure 5, please consider plotting individual data points. This will help the manuscript appear more consistent overall.

Author Response

Dear reviewer, thank you very much for your work; you have helped to make our manuscript better.

Reviewer 3:

Thank you for revising the manuscript. Most of the concerns raised have been addressed. However, there are a few minor things that were either not revised or not explained.

Comments 1:

1. In Figure 3, “μg” is still written as “ug.” Please correct the unit notation.

2. Similar to Figure 5, please consider plotting individual data points. This will help the manuscript appear more consistent overall.

Response 1: We have made the suggested corrections to Figure 3.